# Genetic Parameters for Functional Longevity, Type Traits, and Production in the Serbian Holstein

**DOI:** 10.3390/ani13030534

**Published:** 2023-02-02

**Authors:** Radica Djedović, Natascha Vukasinovic, Dragan Stanojević, Vladan Bogdanović, Hasan Ismael, Dobrila Janković, Nikolija Gligović, Muhamed Brka, Ljuba Štrbac

**Affiliations:** 1Institute of Animal Science, Faculty of Agriculture, University of Belgrade, 11080 Belgrade, Serbia; 2Zoetis Veterinary Medicine Research and Development (VMRD), Kalamazoo, MI 49001, USA; 3Department of Animal Science, Faculty of Agriculture, University of Novi Sad, 21000 Novi Sad, Serbia; 4Department of Animal Science, Faculty of Agriculture and Food, University of Sarajevo, 71000 Sarajevo, Bosnia and Herzegovina

**Keywords:** longevity, type traits, milk traits, Holstein

## Abstract

**Simple Summary:**

Quantitative characteristics of animals such as traits of productivity, reproduction, physical development, longevity, etc. are influenced by many genes, of which each individual gene has a small effect on the overall expression. There are small differences between these genes, and the number of gene combinations in the group is large, which affects the continuous variability of quantitative traits. These traits are also significantly influenced by non-genetic factors that modify their final expression. The evaluation of genetic parameters (heritability and genetic correlations) of quantitative traits is of crucial interest for breeders, and therefore, these parameters are the main imperative in selection. The genetic progress of quantitative traits is achieved by increasing the accuracy of the assessment of genetic variance and selection accuracy, which depends, among other things, on the estimated genetic correlations between the selected traits. The goal of this research was to evaluate genetic variances and covariances of milk yield, longevity, and type traits of the Holstein breed in Serbia in order to assess their role in practical breeding as well as the importance of genetic parameters in setting and revising already existing breeding goals and selection programs.

**Abstract:**

In this study, the authors focused on the evaluation of the genetic parameters of longevity, milk yield traits, and type traits in dairy cattle populations in the Republic of Serbia. The total dataset used consisted of production records and pedigree data for 32,512 Holstein cows that calved from 1981 to 2015. The animal model was applied to determine the variance and covariance components and genetic parameters of the analyzed traits by applying the restricted maximum likelihood (REML) approach and using the programs VCE6 and PEST. The heritability of longevity traits was estimated using the Survival Kit V6.0 software package. Variance and covariance were estimated for five production traits: milk yield (MY), fat yield (FY), protein yield (PY), milk fat content (MC), and protein content (PC), and three longevity traits: length of productive life (LPL), lifetime milk yield (LMY), and the number of lactations achieved (NL) as well as for 18 standard type traits. Heritabilities for the milk, fat, and protein yield traits were 0.20 (MY), 0.15 (FY), and 0.19 (PY), respectively. The estimated coefficients of heritability for the longevity traits were higher when using the Weibull proportional hazards model compared to the traditional linear methods and ranged from 0.08 for NL to 0.10 for LPL. Heritability values for the type traits varied from a low of 0.10 (RLSsv—rear legs set–side view) to medium values of 0.32 (ST-stature). Genetic correlations were found between MY and the following longevity traits: LPL, LMY, and NL with values of −0.18, −0.11, and −0.09, respectively. Genetic correlations were found between MY and a number of linear type traits and varied from 0.02 (between MY and RUH-rear udder height) to 0.28 (between MY and FUA-fore udder attachment). Genetic correlations between the 18 investigated type traits ranged from −0.33 between TL (top line) and RTP (rear teats position) to 0.71 between AN (angularity) and RUH (rear udder height). Genetic correlations between most linear type traits and longevity traits (LPL, LMY, and NL) were generally negative and very low. The highest positive genetic correlation was found between UD and LPL (r_xy_ = 0.38).

## 1. Introduction

Bearing in mind the forecasts that predict the world’s human population will reach 9.7 billion by 2050 [1], the fact that the demand for animal proteins including milk and dairy products will significantly increase is inevitable. Improving milk production through selection will enable breeders to produce larger quantities of milk per animal and unit of input (i.e., feed).

The Holstein cattle breed has the highest milk yield of all dairy breeds in Serbia. It is primarily raised in areas with intensive agricultural production in the vicinity of large cities such as Belgrade and Novi Sad. According to the expert report of the Main Breeding Organization for the region of Central Serbia [2], realized milk production in the breeding herd of Holstein cows in the period from 2007 to 2016, in a standard lactation, was about 7000 kg of milk, with 3.58%, or 247 kg of milk fat. Controlled heifers had a milk protein content of 3.14% (i.e., 217 kg). The stated data on milk yield characteristics are below the average of developed countries of Europe and North America, and the main reasons stated for this are suboptimal environmental conditions, unbalanced nutrition, the use of lower EBV bull semen as well as the privatization of large co-op farms, which has hampered adequate genetic improvement and the management of dairy herds in Serbia.

Planned breeding of Holstein cows in Serbia began in the mid-20th century, mainly on large co-op farms where crossbreeding was often carried out between the already imported Danish Black and White cows and progeny tested Holstein bulls originating from a number of European countries. At that time, breeding heifers and cows were imported mainly from the Netherlands and later also from other European countries (Germany, Denmark, Belgium). The next significant import of Holstein cattle into Serbia was in 1970, when the import of Holstein heifers from the USA began, so that by 1978, a total of 3100 were imported. In addition to the import of heifers, the greatest contribution to genetic progress, especially of the milk yield traits, was achieved by importing the semen of sires for mating with the best cows in order to obtain high-quality male calves that were progeny tested at Serbian national centers for artificial insemination. This program achieved a significant improvement in Black and White cows and the Holstein breed in our country. Next, larger imports of heifers that mainly originated from the Netherlands occurred in 2005 and 2006. Today, it is estimated that the total share of the Holstein breed in Serbia is around 15%.

The selection of sires as well as the cows themselves is traditionally based on the yield of milk, milk fat, and protein. The inclusion of longevity traits and type traits in the selection index as well as the interest of breeders in improving these traits began in the last two decades with the aim to reduce the herd replacement costs, expenses for veterinary interventions, more efficient management, and thus the overall profitability of milk production.

Cattle traits related to longevity are closely related to the survival rate and the period during which the cow generates income [3]. Assessment of the genetic potential of cattle for longevity is extremely important because: (1) it gives animals the opportunity to express their optimal genetic potential; (2) it enables producers to strengthen the selection criteria, and thereby speed up genetic progress (increase voluntary and decrease involuntary culling) [4,5,6,7,8,9,10,11,12,13,14]. Hu et al. [15] concluded that the selection of longevity traits as well as their contribution in relation to other traits in the comprehensive selection index is different for each individual country and population.

Despite these shortcomings, many authors agree that the greatest contribution of including longevity in breeding programs consists of increasing the proportion of older cows with higher yields while simultaneously reducing the proportion of young cows, which will allow older cows to reach their maximum milk yield depending on their age [11,16]. In order to achieve these goals, in modern breeding programs, emphasis is placed on functional conformity and the productive life efficiency of the cow [15], and increasing attention is being paid to constitution, physical development, and type trait characteristics [17]. The importance of breeding cows with good longevity and type traits has never been greater. It was established that deficiencies in type traits lead to lower production and poorer health status of the animal, and therefore to the premature removal of cows from the herd [18,19,20,21,22]. The range of heritability for type traits is wide for the same traits of cows of the same breed by different researchers as well as between different type traits within one or different studies of the same authors. The majority of authors [23,24,25,26,27,28] found medium to moderately high values for heritability of dairy cow type traits in contrast to longevity, which has a low heritability [5,14,29].

The objectives of this study were (1) to estimate the genetic parameters of longevity, milk yield traits, and type traits in dairy cattle populations in the Republic of Serbia, and (2) compare these estimates with the results of authors around the world in order to provide guidelines and emphasize the importance of including these traits in breeding programs to improve the profitability of the milk production of Serbian Holstein cows.

## 2. Materials and Methods

### 2.1. Data Collection

The dataset used consisted of data on the production and pedigree for 32,512 Holstein cows, collected on seven large community farms located in the vicinity of Belgrade (44.8125° N, 20.4612° E) as well as on 2329 smaller farms owned by private farmers on the territory of the province of Vojvodina in the Republic of Serbia (45.2609° N, 19.8319° E). 

Analysis was conducted on cows born from 1981 to 2013 and with first calvings in the period from 1983 to 2015. Cows were descended from 834 sires. Each sire had at last three daughters in the dataset. Genetic links between analyzed animals were established through the sires. A maximum of six generations of ancestors were known in the pedigree file.

Data on cows were collected by the service that kept register records within PK Belgrade (now Aldahra) and the Main Breeding Organization for Livestock, located at the Livestock Department of the Faculty of Agriculture, University of Novi Sad. The downloaded dataset contained a larger number of animals than those analyzed in this research, because during the preparation of the database, all cows without a known date of birth as well as no known calving date were excluded. After that, cows that did not have milk yield results, crossbreed cows as well as cows with incomplete or unknown origin were also excluded from the dataset. In addition, data on the linear evaluations of animals that lacked some of the required information, cows younger or older than the required age at calving at the time of evaluation, cows that were lactating for more than 305 days at the time of evaluation as well as evaluated cows that were not first-calvers were also excluded.

The pedigree file used in the analysis of survival and the evaluation of genetic parameters of the milk yield trait, longevity trait, and type trait contained data on a total of 43,298 animals.

The collected data contained information on milk yield in the first standard lactation–305 days for milk (MY), fat (FY), and protein (PY) as well as fat (FC) and protein (PC) content. Milk yield control was conducted using the AT4 method, which is the referent milk recording method by the International Committee for Animal Recording [30]. Milk traits in standard lactation were calculated using the standard method [30].

In this study, longevity was measured through three traits: length of productive life (LPL), lifetime milk yield (LMY), and number of lactations achieved (NL). The duration of productive life was defined as the time, expressed in days, from the date of first calving to the date of culling or censoring. Lifetime milk yield (LMY) is the amount of milk produced by a cow during her productive life and is expressed in kilograms (kg), while the number of lactations achieved (NL) represents the number of lactations that a cow has achieved during her productive life.

For cows with an unknown or illogical culling date, the date of the end of the last known lactation was used as the culling date. In case of no culling date, the cow was considered culled if the number of days between the beginning of the last known lactation and the end date of the study was more than 500 days. Otherwise, the animal was treated as a censored entry. The share of complete records was 73%, while the censored data made up 27% of the total volume of data.

The set of available data also contained the results for 18 evaluated linear type traits: stature (ST); top line (TL); chest width (CW); body depth (BD); rump position (RP); rump width (RW); angularity (AN); rear legs set, rear view (RLSrv); rear legs set–side view (RLSsv); foot angle (FA); fore udder attachment (FUA); front teats placement (FTP); front teats length (FTL); udder depth (UD); rear udder height (RUH); suspensory ligament (SL); rear teats position (RTP); and rear teats length (RTL). Type traits were evaluated after the first calving (average age of heifers at first calving was 27 months). The average age of animals at the time of evaluation was 30 months (minimum 20, maximum 45). The evaluation was performed according to the ICAR recommendations [31] in the period from 30 days after calving to 210 days of lactation. The achieved average number of days in lactation, on the day of evaluation for heifers for which the data were analyzed, was 95 days. Heifers were linearly evaluated by 22 evaluators, with scores from 1 to 9, within 12 regions. All evaluators were trained according to the Instructions for the Evaluation of Linear Type Traits and Body Development in the Holstein–Friesian breed [32]. 

A descriptive statistical analysis of milk yield traits in the first lactation, longevity traits, and type traits is summarized in Table 1.

### 2.2. Statistical Analysis

The effect of fixed factors on the examined traits was investigated using the GLM procedure within the SAS 9.4 software package [33]. The influence of fixed factors was tested using the step-by-step method, so that the models used in this research only included factors that showed a statistically significant effect within the mentioned procedure. The effect of the individual was included as a random factor.

#### 2.2.1. Animal Model

The animal model was applied to determine the variance and covariance components and genetic parameters of analyzed traits using the restricted maximum likelihood (REML) approach through the VCE6 (Groeneveld et al., 1990) [34] and PEST (Groeneveld et al., 2010) [35] programs.

Analysis and genetic evaluation of the milk production traits in the first standard lactation were performed using the following multi-trait model (Equation (1)):Y_ijklo_ = µ + F_i_ + G_j_ + S_k_ + U_l_ + a_o_ + e_ijklo_,(1)
where Y_ijklo_ is the phenotypic expression of observed milk yield and longevity traits; µ is the average value for the observed trait in the analyzed population; F_i_ is the fixed effect of the farm where the cow produced (i = 2336); G_j_ is the fixed effect of year of first calving (j = 33); S_k_ is the fixed effect of season of first calving (k = 4; winter, spring, summer, autumn); U_l_ is the fixed effect of age at first calving (animals’ age in months): I (19–23), II (24–26), III (27–30), IV (31–33), and V (34–44)); a_o_ is the random effect of the animal; e_ijklo_ is the effect of other factors not included in the model.

Genetic evaluation of the longevity traits was performed using the following model (Equation (2)):Y_ijklmo_ = µ + F_i_ + G_j_ + S_k_ + U_l_ + R_m_ + a_o_ + e_ijklmo_,(2)
where Y_ijklmo_ is the phenotypic expression of the investigated trait; e_ijklmo_ is the random error; µ, F_i_, G_j_, S_k_, U_l_, and a_o_ are the model variables as defined in the previous model (Equation (1)). 

R_m_ is the fixed influence of relative milk yield in the first lactation expressed in standard deviations in relation to the average herd (I x < −1.5 SD; II −1.5 SD < x < −1 SD; III −1 SD < x < −0.5 SD; IV −0.5 SD < x < −0.2 SD; V-0.2 SD < x < 0.2 SD. VI 0.2 SD < x < 0.5 SD; VII 0.5 SD < x < 1 SD; VIII 1 SD < x < 1.5 SD; IX x > 1.5 SD),

Genetic variances and covariances for type traits were evaluated using the following model (Equation (3)):Y_ijlkcmnpo_ = µ + F_i_ + GG_j_ + Y_l_ + S_k_ + U_c_ + O_m_ + Y_n_ + FL_p_ + a_o_ + e_ijlkcmnpo_,(3) where Y_ijlkcmnpo_ is the phenotypic expression of the investigated trait; µ is the general population average; F_i_ is the fixed effect of farm (i = 2336 farms); GG_l_ is the fixed effect of genetic group [interaction of sire’s year of birth (1980–2011) and country of origin (12), 79 genetic groups in total]; Y_l_ is the fixed effect of year of first calving (l = 5 years); S_k_ is the fixed effect of season of first calving (k = 4 seasons (winter, spring, summer, autumn); U_c_ is the fixed effect of age at first calving (animals’ age in months, allocated to five classes: I (19–23), II (24–26), III (27–30), IV (31–33), and V (34–44)); O_m_ is the fixed effect of the evaluator (m = 22); Y_n_ is the fixed effect of the year of evaluation (*n* = 4 years); FL_p_ is the fixed effect of lactation stage at the moment of evaluation (lactation stage in days, allocated to seven classes: I (0–30), II (31–60), III (61–90), IV (91–120), V (121–150), VI (151–180), and VII (181–210)); a_o_ is the random effect of the animal; e_ijlkcmnpo_ is the random error.

The above multi-trait animal model used to estimate the components of variance and covariance can be represented in the form of matrices as follows (Equation (4)):(4)$$\left[\begin{array}{c} y_{1}\\ \vdots \\ y_{n} \end{array}\right]=\left[\begin{array}{ccc} X_{1} & 0 & 0\\ 0 & \ddots & 0\\ 0 & 0 & X_{n} \end{array}\right]\left[\begin{array}{c} b_{1}\\ \vdots \\ b_{n} \end{array}\right]+\left[\begin{array}{ccc} Z_{1} & 0 & 0\\ 0 & \ddots & 0\\ 0 & 0 & Z_{n} \end{array}\right]\left[\begin{array}{c} a_{1}\\ \vdots \\ a_{n} \end{array}\right]+\left[\begin{array}{c} e_{1}\\ \vdots \\ e_{n} \end{array}\right],$$
where *y_i_* is the observation for the i^th^ trait (i = 1 to *n*, where *n* = total number of traits); *b* is the vector of fixed effects; a is the vector of additive genetic effects for the i^th^ trait; e is the vector of random residual effects; X and Z are matrices relating the observations to the fixed and random effects.

The structure of the (co)variances for random effects in the model is the following (Equation (7)):(5)$$Var\left[\begin{array}{c} a_{1}\\ \vdots \\ a_{n}\\ e_{1}\\ \vdots \\ e_{n} \end{array}\right]=\left[\begin{array}{cccccc} g_{1,1}A & \ldots & g_{1,n}A & 0 & \ldots & 0\\ \vdots & \ddots & \vdots & \vdots & \ddots & \vdots \\ g_{n,1}A & \cdots & g_{n,n}A & 0 & \ldots & 0\\ 0 & \ldots & 0 & r_{1,1}I & \ldots & r_{1,n}I\\ \vdots & \ddots & \vdots & \vdots & \ddots & \vdots \\ 0 & \ldots & 0 & r_{n,1}I & \ldots & r_{n,n}I \end{array}\right],$$
where *g* is the additive genetic variance for the direct genetic effect of i^th^ trait (i = *n*-total number of traits); *g_i,j_* is the additive genetic covariance between i^th^ and j^th^ traits; *r_i_*, I is the residual variance for the i^th^ trait; a is the additive genetic relationship matrix; I is the matrix for the residual.

The heritability (*h^2^*) of each trait was computed using Equation (6):(6)$$h^{2}=\frac{\sigma_{a}^{2}}{\sigma_{a}^{2}+\sigma_{e}^{2}},$$
where $\sigma_{a}^{2}$ is the additive genetic variance; $\sigma_{e}^{2}$ is the residual variance; *h^2^* is the heritability.

Genetic correlation (*r_gxy_*) estimates were calculated as follows (Equation (7)):(7)$$r_{gxy}=\frac{Cov_{g(xy)}}{\sqrt{Var_{g(x)}*Var_{g(y)}}},$$
where *Cov_g(xy)_* is the genetic covariance between the *x* and *y* traits; *σ^2^_g(x)_* is the genetic variance of the x trait; *σ^2^_g(y)_* is the genetic variance of the y trait.

#### 2.2.2. Weibull Proportional Hazards Model

Heritability was estimated using survival analysis, with the Survival Kit V6.0 software package (Ducrocq et al., 2010) [36].

The analysis was conducted using the proportional hazards method, and the model itself had the following form (Equation (8)):λ(t) = λ_0_(t) exp (year + season + rpm + farm + uzrpt + lactation + sire)(8)
where λ(t) is the hazard function (current probability of culling) for a particular cow at time t; λ_0_(t) is the Weibull basic hazard function with scalar parameter λ and ρ form parameter; year is the fixed time-dependent effect of year of calving; season is the fixed time-dependent effect of calving season; rpm is the fixed time-dependent influence of relative milk yield within the herd from which the cow originates; farm is the fixed time-independent effect of the farm where the animal was produced; uzrpt is the fixed time-independent influence of age at first calving; lactation is the fixed time-dependent effect of lactation; sire is a random time-independent effect of the sire following a multivariate normal distribution.

Heritability for longevity traits was calculated using the formula proposed by a number of authors [37,38,39], which contains the estimated variance between sires and the proportion of uncensored records. The formula is (Equation (9)):(9)$$h^{2}=\frac{4\sigma_{s}^{2}}{\sigma_{s}^{2}+\frac{1}{p}}$$
where *h*^2^ is the heritability of longevity traits on an ordinal scale; *σ_s_^2^* is the genetic variance between sires; *p* is the share of censored data.

## 3. Results

### 3.1. Phenotypic Variability of Investigated Traits

This research showed that the first standard lactation average yields of MY, FY, and PY were 6644 kg, 231.6 kg, and 200.8 kg, respectively. The average fat content in milk was 3.55%, while the protein content was 3.07%. Longevity trait averages (LPL, LMY, and NL) had the following values: 1292 days, 20,854 kg, and 3.03 lactations, respectively. The lowest average value for type traits was recorded for RTL (4.45), and the highest for AN (6.52).

The stated phenotypic values and variability for the milk yield trait, longevity trait as well as values for 18 linear type traits are presented in Table 1.

### 3.2. Estimates of Variance Components and Heritability

The estimated variance components and the heritabilities of milk production traits are presented in Table 2. The heritability for of milk, fat, and protein yield were 0.20; 0.15 and 0.19, respectively. Higher heritability values were estimated for FC (0.42) and for PC (0.47). Standard errors for the heritability of milk yield traits ranged from 0.018 to 0.025.

Values for the variance components and heritability estimates for the longevity traits obtained with the Weibull proportional hazards model and the linear animal model are presented in Table 3.

Values for heritability of the longevity traits estimated by the Weibull proportional hazards model were higher and amounted to 0.10, 0.09 and 0.08 for LPL, LMY, and NL, respectively, compared to the values estimated using the linear model (0.06, 0.06, 0.07, respectively.

Parameter ρ determined for NL had a lower value compared to the value of the ρ parameters determined for LPL and LMY, which indicates that the risk of culling grew less in the case when longevity was observed through the number of achieved lactations.

Heritability values for the type traits ranged from a low of 0.10 (RLSsv) to a medium value of 0.32 (ST). Additive genetic variance and heritability values are presented in Table 4. The heritability errors for type traits ranged from 0.015 to 0.027. 

### 3.3. Estimates of Genetic Correlations

The genetic correlations between the traits of milk yield and longevity traits are presented in Table 5. Among all of the evaluated genetic correlations, the strongest were between MY and FY (0.99) and between MY and PY (0.96).

Negative correlations were established between MY and the following traits: FC, PC, LPL, LMY, and NL with the following values of −0.15, −0.18, −0.18, −0.11, and −0.09, respectively. All other correlations were positive with standard correlation coefficient errors varying from 0.01 to 0.08. 

Genetic correlations between the milk yield traits and linear type traits are presented in Table 6. The highest positive correlations between the milk yield traits in the first lactation were established for FUA. The estimated values of the genetic correlation coefficients between the linear type traits and milk yield traits in this research indicate the existence of positive correlations between the observed traits, except between the milk yield traits and UD, where the correlations were negative (Table 6). Correlations between UD and MY, FY, and PY were negative and very weak (−0.03, −0.04, and −0.08, respectively). These values indicate that a shallow udder is associated with lower values for MY, FY, and PY yield.

Additionally, several genetic correlations close to zero were established between the analyzed linear type traits, indicating that there is no link between these traits. Genetic correlations between the 18 examined type traits varied from −0.33 between TL and RTP to 0.71 between AN and RUH. Strong positive genetic correlations were found between ST and RW (0.60) as well as between ST and AN (0.69).

Leg and udder traits showed relatively small to moderate genetic correlations within and between groups with one stronger correlation between RUH and SL (0.63). The standard errors for the tested parameters are presented in Table 6 and ranged from 0.01 to 0.08.

Values for the genetic correlations between the type traits and longevity traits are presented in Table 7. In general, genetic correlations between the linear type traits and longevity traits (LPL, LMY, and NL) were negative and very low, except between UD and LPL, where a positive genetic correlation of r_xy_ = 0.38 was found. Values for genetic correlations between LMY and type traits varied from −0.14 (between BD and LMY) to 0.19 (between SL and LMY). Furthermore, the correlations between the type and NL traits were very low, a little below or above zero (Table 7). The standard errors of the genetic correlations between the tested traits presented in Table 7 had values from 0.01 to 0.05.

## 4. Discussion

### 4.1. Phenotypic Variability of Milk Yield, Longevity, and Type Traits

The average milk production in the first standard lactation and milk fat and protein content (MY = 6644 kg, FC = 3.55%, and PC = 3.07%) were in accordance with the values published in studies for first calving in the Holstein and Black and White breeds reared in Serbia at the beginning and during the 2000s [28,40,41,42]. Although there is a potential for high milk production in Serbia, numerous studies carried out in our country on Black and White and Holstein populations show that it manifests itself differently, since the phenotypic expression of milk production is influenced by a large number of factors (genotype, diet, health care, farm management), which are often not in agreement.

The average phenotypic values for milk yield traits in this conducted research (Table 1) were lower than the results obtained in the research by Bohlouli et al. [27] on Iranian Holstein cows, but the obtained values were at the same time higher than the results obtained in Turkey [43,44]. Average yields of milk, fat, and protein in the first lactation in Holstein cows in Poland were 5961.1 kg, 242.9 kg, and 197.6 kg for the yields of milk, milk fat, and protein, respectively (Kruszyński et al. [45]).

The average phenotypic values of the longevity traits in this research indicate that the value for LPL in our populations of Holstein cows was higher compared to the results obtained in Croatia, as presented by Raguž [39]. A study of longevity in a population of Brown Swiss cows by Vukasinovic et al. [46] found that the average value for LPL was 897 days for cows with a known culling date and 1717 days for cows that were treated as incomplete records. When examining the longevity traits of Holstein cows in Spain, Chirinos et al. [47] reported that LPL for Holstein cows in Andalusia was more than 800 days, while the value established for the same trait in Catalonia was more than 900 days. Lower values for LPL were also reported by Nienartowicz-Zdrojewska et al. [48], who examined this trait in a population of Holstein Friesian cows in Poland as well as M’hamdi et al. [49], who analyzed the same trait in a population of Holstein cows in Tunisia. Similar values for LPL were established by Mészáros et al. [38] in the Pinzgauer cow population in Slovakia. Higher values for LPL were established by Cassandro et al. [50], who examined the genetic parameters of the longevity trait in a population of Brown cows in Italy.

Milk production is one of the most important traits in dairy cows, and has a very relevant influence on the decision to cull cows from the herd. Absolute production is not the most adequate selection indicator for making a decision on culling animals from production. A much more reliable indicator is the relative milk production of a cow in relation to the average of the herd in a given year, which is an indicator of a cow’s position in the herd. High-producing cows are significantly less likely to be culled than low-producing cows from the same herd [51,52]. 

When it comes to the type traits of first calvers in this research, the average scores were obtained for the following traits: FTP, FTL, UD as well as for RTP and RTL. The listed traits were close to the average scores for first calvers of the Holstein breed (5), which are also the ideal scores for these traits, according to the ICAR and WHFF nomenclature [31,33]. Regarding the traits of FUA, RUH, and SL, their average values were also closer to the average (5) than the ideal scores (9), according to the nomenclature of ICAR and WHFF [31,32]. The obtained average values for the linear type scores for first calver heifers in the conducted research, in relation to the defined ideal linear scores, deviated the most for the traits ST, CW, and RW, and it is necessary to correct them and pay special attention to their selection in the future. Average values for body traits indicate first calvers of medium development and frame size, with a good slope of the pelvis, which is slightly wider than the average for the HF breed. First calvers of the HF breed in Serbia have a solid frame for dairy cows, but at the same time, there is much room for their further improvement. Zavadilova et al. [19] established that smaller and narrower cows had a lower risk of being culled from the production herd compared to cows with larger frames. Špehar et al. [53] also found that less robust cows remained in the herd longer than cows of moderate size. 

### 4.2. Heritability for Traits of Milk Yield

As one of the most important genetic parameters, heritability helps breeders to determine to what extent genes control the expression of a certain trait, enables the evaluation of the effect of selection as well as helps make a decision as to whether a given trait is more effectively improved through selection or through improving the conditions of housing, care, and health protection of animals.

The estimated heritability of 0.2 for MY in this study (Table 2) showed a moderate value compared to values in literature, with these estimates dependent on several factors related to the volume of the available dataset considered: breed, completeness of the pedigree, model applied, intensity of selection in a given population, etc. The stated value of heritability for MY was lower compared to the results of Bohlouli et al., Kadarmideen and Wegmann, Liu et al., Kudinov et al. and Campos et al. [27,54,55,56,57]. The established heritability for FY of 0.14 was lower than the results published by Campos et al. and DeGroot et al. [57,58], who reported heritability values ranging from 0.20 to 0.24.

Compared to estimates from other studies, the heritability for PY in this paper (0.19) was slightly lower than the values found by Kudinov et al. and DeGroot et al. [56,58], who reported a heritability of 0.20 for PY, which is very close to the result in this paper. For the same trait, Kadarmideen and Wegmann [54] established that the value for heritability was 0.29, while Bohlouli et al. [27] obtained a heritability of 0.31.

The estimated heritabilities for FC and PC (0.42 and 0.47, respectively) were in accordance with the results of Boujenane et al. [59], while the estimated heritability values in this paper suggest a relatively low share of additive genetic variance, especially for the MY and FY traits, which may slow down the genetic progress of these traits in the future.

### 4.3. Heritability for Traits of Longevity

Heritability for traits of longevity in the investigated population was estimated based on formulas presented in Materials and Methods. To assess the heritability of milk yield traits, parameters calculated using the Weibull proportional hazards model were first used.

The estimated variance between sires for LPL was 0.034, resulting in a total additive genetic variance of 0.136. Based on the estimated additive genetic variance and the share of censored records, the value of effective heritability for LPL was calculated as having a value of 0.10.

Significantly higher values for variance between sires in their research were established by Potočnik et al. [60] in the population of Holstein cows in Slovenia as well as by M’hamdi et al. [49] in the population of Holstein cows in Tunisia. Similar values for the variance between sires were established in their research by Raguž [39] in the population of Holstein cows in Croatia as well as by Kernet al. [61] in a population of Holstein cows in Brazil. The heritability value for LPL was between the values published by M’hamdi et al. and Vollema and Groen [49,62]. The established heritability value for LPL (0.10) was almost identical to the value established in the population of Black and White cows in Serbia (Stanojević et al.) [22], as well as Holstein cows in Croatia, Raguž [39]. Approximate heritability values were also established by Imbayarwo-Chikosi et al., Boettcher et al., and Jenko et al. [14,63,64] using the proportional hazards method. A higher heritability value for LPL was established in the research by Najafabadi et al. [65]. In a population of Holstein cows in Iran, they established a heritability for LPL of 0.18. The reason for the difference between the heritability found in our research and the heritability published by Najafabadi et al. [65] is most likely the disregarding of the share of censored records when calculating the heritability on an ordinal scale.

The ρ parameter (basic risk function) for LMY was 2.17, having a lower value compared to the same parameter established when longevity was observed through LPL (ρ = 2.35), which indicates a slightly lower intensity in the increase of the risk of culling from the first calving to culling, if we look at longevity through LMY. A lower value of variance between sires was also established, so heritability also had a slightly lower value (0.09) compared to LPL.

The established value of the ρ parameter indicates that even in the case of NL, the risk increases from the moment of first calving to culling. The ρ parameter established for NL had a lower value compared to the value of the ρ parameter established for LPL and LMY, which means that the risk of culling grew less when longevity was observed through NL. A lower variance value between sires (0.030) was also established in relation to LPL, which resulted in a lower heritability value for NL (0.08).

Heritability values for longevity traits, calculated using the linear animal model (AM), were from 0.06 for LPL and LMY to 0.07 for NL. Higher values by using AM compared to the use of the sire model were established in the research by Raguž [39]. Close values for heritability for LPL were established by Kern et al., Stanojević et al. and Strandberg [21,22,66]. Regarding the LMY, higher heritability values for this trait were established by Vollema and Groen and Hoque and Hodges [62,67] in a population of Black and White cattle in the Netherlands. Lower heritability values for LMY were established by Kern et al. [21] in a population of Holstein Friesian cows in Brazil. The established heritability value for NL in this research was higher than the value established by Vollema and Groen [62], while Jairath et al. [68] established a heritability close to ours for the observed trait. Based on all the data presented, it can generally be concluded that traits of longevity have low heritability, which at the same time means that they are heavily influenced by environmental factors that vary from country to country and from population to population. Therefore, it is necessary to simultaneously study interactions between the traits of longevity and the very environment that surrounds animals during their entire productive life.

In this study, the heritabilities of the longevity traits were estimated using two different methodologies. Heritability for traits of longevity estimated by the Weibull model of proportional hazards was higher compared to the heritability estimated by standard linear models. Although the application of the Weibull model of proportional hazards resulted in higher values of the heritability of longevity traits, survival analysis was based on the sire model, and thus it is not possible to estimate the breeding values for all animals. Even if the linear model may not be most appropriate for all traits, many researchers prefer linear models due to their lower computer requirements. Depending on the amount of data available in the future and breeding goals, either model can be recommended for use in the Serbian Holstein population.

### 4.4. Heritability for Type Traits

The heritability values of type traits measured on first calvers of the Holstein breed in the Republic of Serbia varied from 0.10 for RLSsv to 0.32 for ST, which generally showed higher genetic variability. The estimated heritability value of 0.32 for ST was lower than the value obtained in the study by Campos et al. [57]. A similar value was obtained by Tapki and Guzey [26] and Zink et al. [69]. Lower heritability values for ST, from 0.23 to 0.29, were obtained by Toghiani and Dadpasand et al. [70,71], while higher values were obtained in research by Němcová et al., Zavadilova et al. and Van der Laak et al. [72,73,74].

TL is not a standard type trait that is evaluated in all countries, and it is included as an optional trait in the linear evaluation program for first calvers of the Holstein breed in Serbia. A lower value of 0.09 was obtained by Dadpasand et al. [71] for the Holstein population in Iran. A higher heritability, with a value of 0.22 to 0.31, was obtained by Tapki and Guzey, Bohlouli et al., and Němcová et al. [26,27,72].

The heritability value of 0.15 for CW, which was estimated in this study, was close to the heritability value for the HF populations in Slovenia and Spain as well as the value of 0.18 for the HF population in the Czech Republic, Estonia, Germany, and Canada [75].

For the BD trait in the investigated first calvers, the heritability value was 0.17. An identical value was obtained by Cassandro et al. [50], while approximate values were calculated by Janković et al. and Jagusiak et al. [17,76]. Traits of the pelvis (position and width) were also evaluated within linear type traits. Heritability values for these traits in the population of first calvers in Serbia were 0.16 for RP and 0.18 for RW. The approximate heritability value for RP was obtained by Toghiani [70], while Cassandro et al. [50] obtained a lower value (0.12). A higher heritability value was obtained by Janković et al., Campos et al., and Zavadilova et al. [17,57,73].

The estimated heritability value for RP was close to the heritability value for the HF population in Norway, while in other Interbull member countries, the heritability was higher, ranging from 0.23 in the South Africa to 0.48 in Scandinavian countries [75].

For the RW trait, the estimated heritability value of 0.18 was the closest to the value obtained by Janković et al. and Bohlouli et al. [17,27]. 

In the linear evaluation system, the trait angularity was evaluated as an indicator of the dairy character of cows. The obtained heritability value for the AN trait in first calvers of the HF breed population in Serbia was 0.22, which is close to the value of 0.23 obtained by Toghiani [70] and Campos et al. [57]. Rabbani-Khourasgani et al. [77] obtained a lower heritability value. The heritability values obtained in this research for leg and hoof traits in the population of first calvers of the HF breed in Serbia were 0.16 for the RLSrv trait and 0.10 for RLSsv. The estimated coefficient for RLSrv was close to the coefficient of 0.14 obtained by Van der Laak et al. [74]. Lower heritability ranging from 0.03 to 0.12 were obtained by Zink et al., Zavadilova et al., and Rabbani-Khourasgani et al. [69,73,77].

The heritability for RLSsv is in agreement with the values obtained by Toghiani and Dadpasand et al. [70,71]. The estimated heritability value for the FA trait (0.14) was close to the value published by Janković et al., Tapki and Guzey, and Bohlouli et al. [17,26,27], while lower coefficients were established by the following researchers: Dadpasand et al., Jagusiak et al., and Rabbani-Khourasgani et al. [71,76,77]. A higher heritability of 0.21 was reported by Van der Laak et al. [74]. The estimated heritabilities in this research for udder traits had low to medium values. For the FUA trait, a heritability of 0.18 agreed with the value obtained by Janković et al. [17] and Campos et al. [57], and is identical to the value published by Van der Laak et al. [74]. For the FTP trait, the heritability established in this study was 0.13 and is in accordance with the values obtained by Janković et al. [17] and Dadpasand et al. [71]. A lower coefficient was obtained by Cassandro et al. [50], while the heritability values of other researchers were higher, ranging from 0.22 to 0.44 [26,27,72,74,78].

FTL is also not a standard type trait and was included as an optional trait in the linear evaluation program in Serbia for first calvers of the Holstein breed. The obtained heritability value for FTL was 0.16, and is in agreement with the value obtained by Janković et al. [17] and Dadpasand et al. [71].

For the UD trait, the estimated heritability was 0.15, which is closest to the value of 0.14 obtained by Janković et al. [17]. Lower values were reported by Duru et al. [44] and Liu et al. [55], while the coefficients obtained by other researchers ranged from 0.23 to 0.41 [26,27,57,72,74].

The heritability obtained in this paper for the RUH trait in first calvers of the HF breed was close to the values of 0.10 and 0.12 obtained by Janković et al. [17] and Cassandro et al. [50]. For the SL trait, the estimated heritability was 0.12. Significantly lower coefficients were obtained by Cassandro et al. [50] and Mikhchi et al. [78], while other researchers obtained higher coefficients ranging from 0.18 to 0.26 [26,44,57].

The heritability value of 0.12 for the SL trait in first calvers of the Holstein breed in Serbia was identical to the heritability for the HF population in Hungary, and approximate to the value of 0.10 for HF populations in Ireland and South Korea [75].

Heritability, the RTP trait, were obtained for HF populations in Italy, Portugal, and the USA [75].

For the RTL trait, the estimated heritability in this paper was 0.23, which was close to the values obtained by Janković et al. and Zink et al. [17,69]. Lower coefficients were published by Bohlouli et al., Cassandro et al., and Dadpasand et al. [27,50,71], while higher coefficients were reported by Campos et al., Zavadilova et al., and Rabbani-Khourasgani [57,73,77].

The heritability value of 0.23 for the RTL trait in the population of first calvers of the HF breed in Serbia was identical to the heritability value for the HF population in Hungary, and approximates the value of 0.24 for the HF population in Australia [75].

Heritabilities estimated for all type traits in this study were generally in accordance with those currently used to estimate EBV in the population of Holstein cattle in the Republic of Serbia as well as with the results for most members of Interbull. Standard errors for heritability were low and varied from 0.015 to 0.027. The estimated values of heritabilities for all type traits show that substantial genetic variability exists within the population of Serbian Holsteins, which will enable accurate genetic evaluation, selection, and mating plans to improve type traits in the next generations.

### 4.5. Genetic Correlations between Traits of Milk Yield and Longevity

Lifetime milk yield (LMY) is one of the most important traits for dairy cattle breeders. The short life span and high percentage of cows that are culled result in a significant increase in the production costs. To reduce costs, it is desirable for cows to remain in the herd longer and that they have a high milk yield (Hu et al. [15]).

On 36,663 Slovenian brown cattle, Jenko et al. [64] established that the genetic correlation between LMY and LPL was very strong (0.96 ± 0.008). In our research, this mentioned correlation between LMY and LPL was weaker at 0.63 ± 0.08. A similar result for the estimated genetic correlation between LMY and LPL (0.56) was obtained in a longevity study in a population of Holstein cows in Canada [67].

Studying the relationship between the traits of milk yield and longevity of Holstein cows, Kaupe et al. [79] found that there was a negative genetic correlation between traits of longevity (LPL and LMY) and MY, which is in line with the results of our research. Kaupe et al. [79] also found that longevity had a negative correlation with milk fat (−0.08), and a positive correlation with protein content (0.01). In our research, longevity traits also had a positive genetic correlation with the yield and content of milk fat and protein. Correlation coefficients between LMY established by Weigel et al. [80] were positive for FY and PY, having values of 0.46 and 0.43, respectively.

A recommendation to people working on the selection and breeders of dairy cattle is that when including longevity in the selection index, the effects of all of these traits on the traits of milk yield should be comprehensively analyzed. The short lifespan of dairy cows not only seriously affects productivity, but also hinders the possibility of selection for other traits (Hu et al. [15]). In order to reduce the costs of replacing cows during production, it is important to select cows with excellent milk yield and desirable values for the longevity traits.

### 4.6. Genetic Correlations between Linear Type Traits and Milk Yield Traits

The estimated values of genetic correlations between the MY and type traits in this study were negative for FA, UD, and RTP (−0.05, −0.03, and −0.02, respectively). These results do not agree with the results established by Bohlouli et al. [27] and Berry et al. [81], who established positive correlations between these traits. In their research on the relationship between udder traits and milk yield, Bohlouli et al. [27] established positive correlations for all traits, ranging from 0.02 for FTP to 0.26 for AN.

In studies by Van Niekerk et al. [82] and DeGroot et al. [58], the correlations between UD and yield traits (MY, MY and PY) were negative and consistent with the results in this study. The genetic correlation established by Harris et al. [83] between UD and MY in the Jersey breed was negative and weak, and at the same time, in agreement with the results in this study. When examining the relationship between udder traits and milk yield of first calvers, Short and Lawlor [84] obtained moderate genetic correlations, with values from −0.48 for UD to +0.54 for AN, while Kruszyński et al. [45] obtained values for genetic correlations ranging from −0.09 for FTP to −0.30 for UD.

In the conducted research, positive genetic correlations between MY and type traits ranged from 0.02 for RUH to 0.28 for FUA. In the Jersey breed of cattle in South Africa, Van Niekerk et al. [82] saw that a positive genetic correlation between MY and FUA was established. Estimated genetic correlations between FUA and MY and PY in the study by these authors were 0.50 and 0.51, respectively, and thus higher than the genetic correlations in this study. The results in this paper suggest that the selection for udder traits would have a positive effect on increasing the milk yield of Holstein cows in Serbia. Evaluated genetic correlations also indicate that the improvement in leg and hock traits could have certain positive effects on the milk yield traits.

### 4.7. Genetic Correlations between Type Traits

Genetic correlations between the analyzed type traits ranged from −0.33 (between TL and RTP) to 0.71 (between AN and RUH). A weak correlation (0.01) existed between TL and FTL as well as between RP and RW, while significantly stronger correlations were recorded between RUH and SL (0.63). Additionally, there was a strong genetic correlation between ST and AN (0.69). Comparing the obtained values for genetic correlations between type traits with other results showed that genetic correlations between FUA and FTP were lower than the values established by Duru et al. [44] in Turkey and Ptak et al. [25] in Iran.

FTP has a weak genetic correlation with other udder traits, which was also confirmed in research by Duru et al. [44]. Genetic correlations for other type traits were also weak such as between FTL and RP (0.12), or moderate such as between BD and RUH (0.36). The genetic correlation between FUA and UD was 0.45 in our study, similar to Duru et al. [44]. Van Niekerk et al. [82], for the Jersey breed in South Africa, established that the genetic correlation between FUA and UD was weak at 0.23, which deviated from the result obtained in this study. However, studies by Meyer et al. [85] and Gengler et al. [86] found moderate positive correlations between FUA and UD (0.44 and 0.48, respectively).

### 4.8. Genetic Correlations between Type Traits and Longevity Traits 

The importance of type traits for assessing longevity traits lies in the fact that type traits are recorded as a mandatory selection measure at a relatively early age, already during the first lactation. At the same time, type traits generally have a medium to moderately high heritability, as opposed to longevity traits, which have a low heritability, making it difficult to increase the genetic progress for these traits [29]. 

Of all the assessed genetic correlations, the genetic correlations between type traits and longevity traits varied the least. The research results in this study were very close to the results published by Vukasinovic et al. [11]. When examining the possibility of using type traits as an additional source of information to increase the accuracy when assessing the breeding value of Swiss Simmental and Red and White cows for longevity traits, the mentioned authors found negative genetic correlations for traits of frame. As for udder traits, these researchers obtained positive very weak to moderate genetic correlations for all traits, except for FTL, which was moderate and negative (−0.41). The same authors confirmed that longevity is genetically related to leg traits and udder traits. Cows having good angularity (AN) have both properly set legs and hocks and well-shaped udders, with genetic correlations between angularity and udder traits ranging from 0.35 to 0.75. Vukasinovic et al. [11] simultaneously confirmed that the reliability of combined selection was higher than the reliability of direct selection alone. The greatest increase in reliability could be seen in young bulls with censored data for their daughters.

Unlike our research, Strapak et al. [18] obtained extremely weak positive correlations between functional longevity and almost all traits of frame (from 0.03 TL to 0.14 for AN). The similarity between our research and the research of Strapak et al. [18] is reflected in the close values for genetic correlations between the longevity traits and udder traits (from 0.15 for RTP to 0.19 for FUA).

The association between LPL and type traits was also investigated by Wasana et al. [87], who obtained negative correlations between all traits of frame including dairy character. The research of Noskova et al. [88] established that the greatest influence on the number of achieved lactations (NL) was exerted by UD, RLSrv, RLSsv, FTL, and RTP.

In a study conducted by Imbayarwo-Chikosi et al. [14], all of the examined type traits, except for the position of the hind legs, side view, showed a significant effect on functional longevity, and can be used as longevity indicators.

Schneider et al. [89], when examining the association between type traits and functional longevity in a Holstein population in Quebec via survival analysis, found that the general linear score and udder traits had the greatest influence on the hazard function. Such results are not unexpected, if it is taken into account that cows are selected for high milk production over a long period of time.

Genetic correlations between the level of physical development and longevity traits in the presented studies showed the importance of type traits in the selection process. It is obvious that the selection for LMY has a positive effect on all udder traits, except for udder depth. LPL is an important and complex longevity trait because it includes many factors: production technology, herd management, reproduction, health status, economic justification of milk production, and also indicates the genetic potential of the analyzed cows.

The results obtained in this study were generally close to the results from studies of other Holstein populations in other countries. Certain desirable correlations were confirmed, the connection between the udder characteristics and longevity characteristics being especially significant.

The results confirm that certain type traits can be included in a selection index, along or instead of the longevity traits to provide additional information on the animals’ genetic merit for longevity, when the longevity phenotype is missing or censored. Using indicator traits can improve the accuracy of genetic evaluation and selection for correlated traits that are expensive and difficult to measure or are recorded relatively late in life.

## 5. Conclusions

The estimated heritability and genetic correlations for the traits of milk yield, longevity, and type of Serbian Holstein suggest their sufficient manifestation and are therefore acceptable for inclusion in the selection index. Given that a large number of traits were investigated, for a future breeding program, and based on the estimated values of genetic variances and covariances, traits that will ensure that cows have a high milk yield with a satisfactory protein content and possess desirable longevity and type traits should be selected. It is also necessary to take into account the desirable and undesirable correlations between the investigated traits in order to ensure economical milk production in the long term.

The estimated genetic parameters of the analyzed traits were compared with a large number of references in order to more objectively evaluate the place of the Serbian Holstein in relation to other countries worldwide.

The values of the genetic parameters of the investigated traits of milk yield, longevity, and type traits in the population of Holstein cows in Serbia were mostly lower or similar compared to the data of other researchers. Given that genetic parameters of quantitative traits depend on the variability of the observed traits and previous selection, lower values of these parameters may be the result of lower or higher genetic variability, caused, among other things, by the influence of systematic factors. The estimated genetic parameters of the analyzed traits are significant for the evaluation of their breeding values, which will be the subject of our future scientific work.

## Figures and Tables

**Table 1 animals-13-00534-t001:** Traits analyzed: abbreviation, units of measurement, ideal score, and descriptive statistics for milk trait, longevity traits, and type traits.

Traits	Abbreviations	Units of Measurement/ Ideal Scores	*n*	Mean	SD	Min	Max	CV (%)
Milk traits								
Milk Yield, 305 d	MY	kg	32,512	6644	1622	1532	14,549	24.41
Milk fat content, 305 d	FC	%		3.55	0.31	1.04	5.02	8.70
Fat Yield, 305 d	FY	kg		231.6	54.38	56.6	492.7	23.44
Milk protein content, 305 d	PC	%		3.07	0.48	0.97	3.95	15.64
Protein Yield, 305 d	PY	kg		200.8	44.2	46.3	378.6	22.10
Longevity traits								
Length of productive life	LPL	days	15,894	1292	764.22	14	4271	59.15
Lifetime milk yield	LMY	kg		20,854	13,653.08	1708	99,816	65.46
Number of lactation	NL	number of lactations achieved by cow		3.03	1.82	1	8	60.06
Type traits								
Stature	ST	8	24,103	5.94	1.60	1	9	26.8
Top line	TL	7		6.19	1.07	1	9	17.2
Chest width	CW	7		5.91	1.21	1	9	20.5
Body depth	BD	7		6.18	1.22	1	9	19.6
Rump position	RP	5		5.88	1.24	1	9	21.2
Rump width	RW	8		6.17	1.20	1	9	19.3
Angularity	AN	9		6.52	1.36	1	9	20.9
Rear legs set–rear view	RLSrv	8		6.30	1.37	1	9	21.6
Rear legs set–side view	RLSsv	5		5.14	1.06	1	9	20.5
Foot angle	FA	7		4.78	1.14	1	9	24.0
Fore udder attachment	FUA	9		5.75	1.41	1	9	24.3
Front teats placement	FTP	6		4.99	1.14	1	9	22.8
Front teats length	FTL	5		5.15	1.10	1	9	21.3
Udder depth	UD	5		5.91	1.18	1	9	21.0
Rear udder height	RUH	9		6.28	1.33	1	9	21.1
Suspensory ligament	SL	9		5.98	1.41	1	9	23.7
Rear teats position	RTP	5		5.73	1.32	1	9	23.1
Rear teats length	RTL	5		4.45	1.10	1	9	24.8

**Table 2 animals-13-00534-t002:** Estimates for the additive genetic variance (*σ^2^_a_*), heritability (*h^2^*), and standard error (SEh^2^) for milk traits.

Type Traits	Animal Model			
	$\sigma_{a}^{2}$	$\sigma_{e}^{2}$	*h* ^2^	SEh^2^
MY	423,155.353	1,690,630.474	0.20	±0.020
FC	0.0514	0.0698	0.42	±0.025
FY	855.645	980.408	0.15	±0.017
PC	0.0101	0.0115	0.47	±0.026
PY	438.067	1842.581	0.19	±0.019

*σ^2^_a_*—additive genetic variance; *σ^2^_e_*—residual variance; *h^2^*—heritability; SEh^2^—standard error for heritability; MY—milk yield; FC—milk fat content; FY—fat yield; PC—milk protein content; PY—protein yield.

**Table 3 animals-13-00534-t003:** Estimates for the additive genetic variance (*σ^2^_a_*), heritability (*h^2^*), and standard error (SEh^2^) for the longevity traits.

Traits	Weibull Proportional Hazards Model					Animal Model			
	ρ	$\sigma_{s}^{2}$	$4\sigma_{s}^{2}$	*h^2^*	SEh^2^	*σ^2^_a_*	*σ^2^_e_*	*h^2^*	SEh^2^
LPL	2.35	0.036	0.144	0.10	0.026	25650.1	408,511	0.06	0.011
LMY	2.17	0.032	0.128	0.09	0.025	9,092,950	14,340,000	0.06	0.011
NL	1.93	0.030	0.120	0.08	0.026	0.162	2.32	0.07	0.012

Ρ—basic risk function; *σ_s_^2^*—genetic variance between sires; *σ^2^_a_* (4*σ_s_^2^*)—additive genetic variance; *σ^2^_e_*—residual variance; SEh^2^–standard error for heritability; LPL—length of productive life; LMY—lifetime milk yield; NL—number of lactation.

**Table 4 animals-13-00534-t004:** Estimates for the additive genetic variance (*σ^2^_a_*), heritability (*h^2^*), and standard error (SEh^2^) for type traits.

Type Traits	Animal Model			
	$\sigma_{a}^{2}$	$\sigma_{e}^{2}$	*h* ^2^	SEh^2^
ST	0.689	1.482	0.32	0.027
TL	0.136	0.920	0.13	0.019
CW	0.176	0.973	0.15	0.021
BD	0.193	0.921	0.17	0.021
RP	0.211	1.092	0.16	0.021
RW	0.214	0.991	0.18	0.023
AN	0.369	1.199	0.22	0.024
RLSrv	0,241	1.283	0.16	0.021
RLSsv	0.107	0.924	0.10	0.015
FA	0.156	0.991	0.14	0.020
FUA	0.306	1.364	0.18	0.024
FTP	0.157	1.048	0.13	0.020
FTL	0.173	0.911	0.16	0.021
UD	0.186	1.221	0.15	0.021
RUH	0.157	1.375	0.11	0.017
SL	0.195	1.329	0.13	0.020
RTP	0.258	1.261	0.17	0.021
RTL	0.363	1.235	0.23	0.025

*σ^2^_a_*—additive genetic variance; *σ^2^_e_*—residual variance; *h^2^*—heritability; SEh^2^—standard error for heritability; ST—stature; TL—top line; CW—chest width; BD—body depth; RP—rump position; RW—rump width; AN—angularity; RLSrv—rear legs set–rear view; RLSsv—rear legs set–side view; FA—foot angle; FUA—fore udder attachment; FTP—front teats placement; FTL—front teats length; UD—udder depth; RUH—rear udder height; SL—suspensory ligament; RTP—rear teats position; RTL—rear teats length.

**Table 5 animals-13-00534-t005:** Genetic correlations (r_xy_) between the milk traits and longevity traits.

Traits	2 FC	3 FY	4 PC	5 PY	6 LPL	7 LMY	NL
1 MY	−0.15 ± 0.02	0.99 ± 0.06	−0.18 ± 0.04	0.96 ± 0.05	−0.18 ± 0.05	−0.11 ± 0.02	−0.09 ± 0.02
2 FC	-	0.98 ± 0.08	0.19 ± 0.05	0.21 ± 0.04	0.19 ± 0.04	0.16 ± 0.03	0.05 ± 0.01
3 FY	-	-	0.20 ± 0.04	0.22 ± 0.04	0.20 ± 0.05	0.16 ± 0.02	0.06 ± 0.01
4 PC	-	-	-	0.99 ± 0.08	0.25 ± 0.04	0.14 ± 0.03	0.10 ± 0.02
5 PY	-	-	-	-	0.27 ± 0.04	0.16 ± 0.04	0.16 ± 0.04
6 LPL	-	-	-	-	-	0.63 ± 0.08	0.55 ± 0.07
7 LMY	-	-	-	-	-	-	0.71 ± 0.07

MY—milk yield; FC—milk fat content; FY—fat yield; PC—milk protein content; PY—protein yield; LPL—length of productive life; LMY—lifetime milk yield; NL—number of lactation.

**Table 6 animals-13-00534-t006:** Genetic correlations (r_xy_) between the milk traits and type traits.

Traits	6 ST	7 TL	8 CW	9 BD	10 RP	11 RW	12 AN	13 RLSrv	14 RLSsv	15 FA	16 FUA	17 FTP	18 FTL	19 UD	20 RUH	21 SL	22 RTP	23 RTL
1 MY	0.09	0.18	0.04	0.22	0.08	0.06	0.22	0.05	0.04	−0.05	0.28	0.09	0.08	−0.03	0.02	0.05	−0.02	0.08
2 FC	0.00	−0.13	0.00	0.05	−0.07	−0.03	0.18	0.00	0.00	0.01	0.05	0.12	0.03	−0.02	0.12	0.02	0.00	0.01
3 FY	0.10	0.18	0.04	0.26	0.03	0.05	0.24	0.05	0.05	−0.07	0.23	0.22	0.11	−0.04	0.06	0.10	0.04	0.07
4 PC	0.00	0.08	0.03	0.04	−0.02	−0.12	0.05	0.02	0.01	0.00	0.19	0.15	0.04	−0.09	0.18	0.05	0.01	0.02
5 PY	0.10	0.22	0.06	0.06	0.08	0.01	0.22	0.07	0.06	−0.05	0.31	0.10	0.02	−0.08	0.07	0.13	0.03	0.08
6 ST	-	0.21	0.17	0.52	−0.08	0.60	0.69	0.26	0.22	0.27	0.30	0.05	0.11	0.32	0.29	0.24	0.12	0.06
7 TL		-	0.22	0.27	−0.18	−0.19	0.14	0.04	−0.05	0.00	0.22	0.00	0.01	0.11	0.26	−0.03	−0.33	−0.14
8 CW			-	0.54	−0.07	0.41	−0.23	0.32	−0.19	0.22	0.20	0.15	0.00	0.09	0.27	−0.18	−0.26	−0.11
9 BD				-	−0.02	0.51	0.01	0.01	−0.06	0.18	0.15	−0.03	0.10	0.00	0.36	0.05	0.15	0.05
10 RP					-	0.10	0.11	0.04	−0.15	0.02	0.21	0.15	0.12	0.23	0.33	−0.24	−0.01	−0.03
11 RW						-	−0.21	0.16	0.18	−0.01	0.18	0.00	0.14	0.15	0.42	0.00	0.09	0.00
12 AN							-	0.19	0.14	0.02	0.16	−0.10	0.38	−0.12	0.71	0.52	0.19	0.06
13 RLSrv								-	0.54	−0.15	0.34	0.17	0.12	0.07	0.11	0.35	0.16	0.17
14 RLSsv									-	0.12	0.39	0.19	0.08	0.00	0.17	0.27	0.18	0.05
15 FA										-	0.00	−0.05	0.01	0.16	0.14	−0.31	−0.13	0.04
16 FUA											-	−0.13	0.22	0.45	0.30	0.37	0.47	0.23
17 FTP												-	−0.02	−0.01	−0.12	0.02	0.32	0.14
18 FTL													-	0.14	0.08	0.03	−0.11	−0.06
19 UD														-	0.00	0.14	0.16	0.19
20 RUH															-	0.63	0.53	0.37
21 SL																-	0.48	0.31
22 RTP																	-	0.40
23 RTL																		-

MY—milk yield; FC—milk fat content; FY—fat yield; PC—milk protein content; PY—protein yield; ST—stature; TL—top line; CW—chest width; BD—body depth; RP—rump position; RW—rump width; AN—angularity; RLSrv—rear legs set–rear view; RLSsv—rear legs set–side view; FA—foot angle; FUA—fore udder attachment; FTP—front teats placement; FTL—front teats length; UD—udder depth; RUH—rear udder height; SL—suspensory ligament; RTP—rear teats position; RTL—rear teats length; Standard errors for the parameters were in the range from 0.010 to 0.08. Positive genetic correlations were found between MY and a number of linear type traits and they varied from 0.02 (between MY and RUH) to 0.28 (between MY and FUA).

**Table 7 animals-13-00534-t007:** Genetic correlations (r_xy_) between the longevity traits and type traits.

Type Traits	LPL	LMY	NL
1 ST	−0.27	0.05	−0.11
2 TL	−0.18	−0.11	−0.14
3 CW	−0.27	−0.02	−0.05
4 BD	−0.08	−0.14	0.08
5 RP	−0.10	0.01	0.04
6 RW	−0.14	−0.01	−0.09
7 AN	−0.16	0.07	−0.17
8 RLSrv	0.01	0.02	0.03
9 RLSsv	0.13	−0.05	0.01
1 0FA	−0.12	0.06	−0.02
11 FUA	0.17	0.02	−0.03
12 FTP	0.10	0.04	0.01
13 FTL	0.08	0.09	0.08
14 UD	0.38	0.16	0.15
15 RUH	0.06	0.02	0.06
16 SL	0.15	0.19	0.03
17 RTP	0.11	0.05	0.07
18 RTL	−0.04	0.03	0.01

ST—stature; TL—top line; CW—chest width; BD—body depth; RP—rump position; RW—rump width; AN—angularity; RLSrv—rear legs set–rear view; RLSsv—rear legs set–side view; FA—foot angle; FUA—fore udder attachment; FTP—front teats placement; FTL—front teats length; UD—udder depth; RUH—rear udder height; SL—suspensory ligament; RTP—rear teats position; RTL—rear teats length; LPL—length of productive life; LMY—lifetime milk yield; NL—number of lactation. Standard errors for the parameters were in the range from 0.010 to 0.05.

## Data Availability

The original data used in this article are available by contacting the corresponding author upon request.

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
