# Peer review of "Genetic Parameters for Functional Longevity, Type Traits, and Production in the Serbian Holstein"

_animals, 2023, doi:10.3390/ani13030534_

Round 1
Reviewer 1 Report
1.Whether there is only AT4 method to control milk collection? What are the advantages and disadvantages of this method compared with other methods.
2.The average age of cows used for experimental data collection is 30 months. Is there any basis? How does too young or too old age affect the experimental data?
3.What is the reason for the difference between the estimates of Weibull proportional hazards model and linear animal model?
4. How to correct the deviation between the mean value of the linear type score and the ideal score of the newborn cow?
5. How to minimize the risk from the first calving to culling?
6.How to evaluate the relationship between the traits of dairy cows and their likelihood of being eliminated?
7.Apart from the skeleton, what other traits can reduce the risk of elimination of dairy cows?
8.How the relatively low sexual genetic variance slows down the future genetic process of traits?
9. References in the article are not superscripted
10. Too many spaces on table7
11. 16 pages of references, e.g. 51, et al followed by no period and elsewhere
Author Response
Reviewer 1:
As authors of the paper "Genetic Parameters for Functional Longevity, Type Traits and Production in the Serbian Holstein", we very much appreciate your effort to evaluate our manuscript extremely carefully and very expertly and professionally.
1.Whether there is only AT4 method to control milk collection? What are the advantages and disadvantages of this method compared with other methods?
The referent milk recording method by the International Committee for Animal Recording (ICAR, 2003) is the A4 method. The reference has been added (line 173-174). There are also others methods (AT4, A6, and AT6) which are modified variants. The A4 method is the most complete method, but also the most expensive. Varieties of this method (AT4, A6, AT6) exist primarily for more practical reasons and lower costs. According to Ganter et al. (2008), 100-day and 200-day milk yields predicted from AT4, A6 and AT6 milk recording methods did not differ significantly from those predicted using the A4 method.
The advantage of modified variants is in the satisfactory accuracy at an acceptable price.
2.The average age of cows used for experimental data collection is 30 months. Is there any basis? How does too young or too old age affect the experimental data?
According to the instructions for the linear evaluation of Holstein first-calf cows in Serbia, which is in accordance with the recommendations of ICAR, it is recomended that the evaluation should be done 210 days from the moment of the first calving (7 months). Considering that cows first calve at the age of about 24 months, we arrive at the moment of evaluation at the age of about 30 months. The age of animals in our study was limited to 24 to 45 months (line 196) in order to avoid bias that could have been caused by evaluation too young or too old animals.
3.What is the reason for the difference between the estimates of Weibull proportional hazards model and linear animal model?
The Weibull proportional hazards model more adequately describes the non-normally distributed nature of longevity data. The Weibull method of proportional hazards allows the inclusion of time-dependent variables as well as data on the longevity of cows that are still in production (censored records), which greatly contributes to a more precise assessment of the additive genetic variance of longevity traits. On the other hand, the linear model consideres observed values of longevity data without completely taking into account underlying risk - therefore the differences in the parameter estimates. This is explained in the paper in lines 563-568. However, a linear model may be preferred in some situations (large amount of data coupled with less than optimal computational resources) due to its easier convergence and lower computational requirements (mentioned in lines 568-569).
- How to correct the deviation between the mean value of the linear type score and the ideal score of the newborn cow?
We will do planned mating where we will use bulls with high breeding values for traits that have a significant deviation from the type score and thus improve the expression of those traits in the next generation. This will be possible due to relatively high heritabilities of type traits, and has been added to the manuscript (lines 695-698).
- How to minimize the risk from the first calving to culling?
Longevity traits are traits with low heritability, so the influence of external environmental factors is key to the phenotypic expression of this group of traits. In order to reduce the risk of culling first-calf cows, they should calve at an optimal age (older cows at first calving have a higher risk of culling). Also, special attention should be paid to health and reproduction, because these are the dominant reasons for culling of first-calf cows. Traits correlated with longetvity can be included in a selection index, as indicated in lines 861-865.
- How to evaluate the relationship between the traits of dairy cows and their likelihood of being eliminated?
This relats to your previous question. We can evaluate the relationship between dairy cow traits and the risk of culling in two ways. The first way is by calculating the genetic correlation between milk, type or reproduction traits and longevity traits, most often using linear models. The second way is within the survival analysis, where we will include the milk traits, type or other traits as indicators of longevity traits and get culling risks depending on their expression. It is necessary to take into account desirable and undesirable correlations between traits in order to ensure economical milk production in the long term (lines 865-866 in the paper).
7.Apart from the skeleton, what other traits can reduce the risk of elimination of dairy cows?
Good performance of health and resistance traits, as well as reproductive traits, have a key contribution to reducing the risk of culling. The correlations between all investigated type and longevity traits are presented in Table 7 in the paper.
8.How the relatively low sexual genetic variance slows down the future genetic process of traits?
High rate of unwanted culling of dairy cows (most culling from production are unwanted, up to 90% in some dairy cattle populations) reduce the intensity of selection in the female sex.
- References in the article are not superscripted
References are done in accordance to the journals instructions.
- Too many spaces on table7
Corrected
- 16 pages of references, e.g. 51, et al followed by no period and elsewhere
Corrected and checked throughout the manuscript.

Reviewer 2 Report
This paper reported heritability and genetic correlations for three groups of traits in the Serbian Holstein. I think it was well-written and is an important summary of those genetic parameters which can be used in the genetic evaluation for Serbian Holstein. I have a few suggestions below:
1) In the 2.2 Statistical Analysis session, I feel some paragraphs could be rearranged to make it clearer or add some sub-titles. you can talk about linear animal models for all three groups of traits first, followed by h2 and genetic correlation Equations. Then you can talk about additional survival analysis for longevity traits.
2) As described in the paper, a multi-trait animal model was used in this study; did you mean all three groups of traits were analyzed in one run? or did you mean a series of bivariate analyses? If all traits were run altogether, were there any convergence issues?
3) The terms 'coefficient of heritability' and 'heritability' were used. I think it is better to be consistent with one term only throughout the paper.
4) The term 'standard heritability error' is not commonly used. I prefer ' standard error for heritability'.
5) in Tables 2 and 4, 'Animal' should be 'Animal Model.'
6) In Lines 299 and 307, 'standard error' should be 'standard error for heritability.'
7) Line 357 belongs to the footnote of Table 6?
8) Line 367, Standard errors are not presented in Table 6; is that correct?
9) The small genetic correlation estimates might not be statistically significant from 0 if the standard error of those estimates were large. So, you should be careful to make statements like negative or positive correlations between two traits.
Author Response
Reviewer 2:
As authors of the paper "Genetic Parameters for Functional Longevity, Type Traits and Production in the Serbian Holstein", we very much appreciate your effort to evaluate our manuscript extremely carefully and very expertly and professionally.
This paper reported heritability and genetic correlations for three groups of traits in the Serbian Holstein. I think it was well-written and is an important summary of those genetic parameters which can be used in the genetic evaluation for Serbian Holstein. I have a few suggestions below:
1) In the 2.2 Statistical Analysis session, I feel some paragraphs could be rearranged to make it clearer or add some sub-titles. you can talk about linear animal models for all three groups of traits first, followed by h2 and genetic correlation Equations. Then you can talk about additional survival analysis for longevity traits.
As recommended the 2.2 Statistical Analysis section was rearranged and sub-titles were added.
2) As described in the paper, a multi-trait animal model was used in this study; did you mean all three groups of traits were analyzed in one run? or did you mean a series of bivariate analyses? If all traits were run altogether, were there any convergence issues?
Analyses were done in multiple runs (in different stages), depending on the goal of the analysis and the traits involved, which can be seen from models presented under Materials and Methods.
3) The terms 'coefficient of heritability' and 'heritability' were used. I think it is better to be consistent with one term only throughout the paper.
Corrected throughout the manuscript
4) The term 'standard heritability error' is not commonly used. I prefer ' standard error for heritability'.
Corrected throughout the manuscript
5) in Tables 2 and 4, 'Animal' should be 'Animal Model.'
Corrected
6) In Lines 299 and 307, 'standard error' should be 'standard error for heritability.'
Corrected
7) Line 357 belongs to the footnote of Table 6?
This line is part of the footnote and has been corrected.
8) Line 367, Standard errors are not presented in Table 6; is that correct?
Standard errors are now in the footnote. This was done for clarity reasons due to tables size.
9) The small genetic correlation estimates might not be statistically significant from 0 if the standard error of those estimates were large. So, you should be careful to make statements like negative or positive correlations between two traits.
Thank you for the suggestion regarding the error values ​​of genetic correlations, which are particularly high in table 7, as a result we were more careful in the manuscript when emphasizing positive and negative correlations.

Reviewer 3 Report
In this study, the heritability of several traits and milk quality traits of cows in Serbia were analyzed, and the correlations between milk traits and the type traits of cattle were performed. Among them, several traits related to the longevity of cows were focused on and evaluated. This study provides valuable information for dairy cattle genetics and breeding. However, this manuscript is written in a form more like a dissertation, and further streamlining of language is recommended, along with a revised format of all tables.
1. I remember the journal has a word limit for the Abstract, so please summarize this section for approval of the limitations.
2. The Introduction does not need to provide so much information on the importance of cow longevity. It is suggested that be reduced number of words to present better the current state of research on trait and genomic inheritance in dairy cows worldwide.
3. Line122 change “paper” to “study”.
4. Please give information on how the cows were raised and the nutritional conditions. It is recommended that background information such as breeding goals and plans (Lines 123-137) be presented in the INTRODUCTION or supplemental file.
5. The authors cite a large amount of data from other researchers in their discussion, some of which are simply comparative, and suggest adding some summary and speculative conclusions to clarify the possible reasons for these results.
6. The discussion section is a bit too long and can be reduced.
7. The conclusion does not summarize well the main results of this work. It needs to be rewritten considering the key results of this work.
8. Line 348 please revise "See Table 6" to “Table 6”.
9. Line 374 please revise "See Table 6" to “Table 7”.
Author Response
Reviewer 3:
As authors of the paper "Genetic Parameters for Functional Longevity, Type Traits and Production in the Serbian Holstein", we very much appreciate your effort to evaluate our manuscript extremely carefully and very expertly and professionally.
In this study, the heritability of several traits and milk quality traits of cows in Serbia were analyzed, and the correlations between milk traits and the type traits of cattle were performed. Among them, several traits related to the longevity of cows were focused on and evaluated. This study provides valuable information for dairy cattle genetics and breeding. However, this manuscript is written in a form more like a dissertation, and further streamlining of language is recommended, along with a revised format of all tables.
Precisely because of the large number of analysed traits (a total of 26 traits; 5 production traits; 3 longevity traits and 18 standard type traits) and the assessment of their genetic parameters, as well as the desire to compare obtained results with numerous results of authors from all over the world, and to thus create a realistic picture of the place that the Serbian Holstein population holds according to its genetic potential, in relation to other countries, in order to create the most objective and modern breeding program for this breed, the total scope of the paper, and especially the discussion, are quite extensive.
- I remember the journal has a word limit for the Abstract, so please summarize this section for approval of the limitations.
According to your esteemed recommendation and the recommendation of Animals journal, the Abstract of this paper has been shortened.
- The Introduction does not need to provide so much information on the importance of cow longevity. It is suggested that be reduced number of words to present better the current state of research on trait and genomic inheritance in dairy cows worldwide.
We also thank you for this suggestion and have shortened the Introduction by removing this information.
- Line122 change “paper” to “study”.
Corrected
- Please give information on how the cows were raised and the nutritional conditions. It is recommended that background information such as breeding goals and plans (Lines 123-137) be presented in the INTRODUCTION or supplemental file.
This information was moved to the introduction as you requested.
- The authors cite a large amount of data from other researchers in their discussion, some of which are simply comparative, and suggest adding some summary and speculative conclusions to clarify the possible reasons for these results.
We also adopted this comment. We added a short explanation to the conclusion to address these results. This addendum now reads:
“Estimated genetic parameters of the analysed traits were compared with a large number of references in order to more objectively evaluate the place of Serbian Holstein in relation to other countries in the world, all with the aim of evaluating their breeding values.”
- The discussion section is a bit too long and can be reduced.
We agree that part of the discussion should be shortened and it was reduced, especially in the section "Heritability for Type Traits and Genetic Correlations between Type Traits and Longevity Traits" dealing with the majority of traits.
- The conclusion does not summarize well the main results of this work. It needs to be rewritten considering the key results of this work.
The conclusion has been completely rewritten and we hope it better summarizes the results of this study.
- Line 348 please revise "See Table 6" to “Table 6”.
Corrected
- Line 374 please revise "See Table 6" to “Table 7”.
Corrected
